# Does Parity Influence the Magnitude of the Stress Response of Nellore Cows at Weaning?

**DOI:** 10.3390/ani13081321

**Published:** 2023-04-12

**Authors:** Camila de Paula, Luciana Navajas Rennó, Matheus Fellipe de Lana Ferreira, Samira Silveira Moreira, Hudson Caio Martins, Isabela Iria Rodrigues, Edenio Detmann, Sebastião de Campos Valadares Filho, Mário Fonseca Paulino

**Affiliations:** 1Department of Animal Science, Universidade Federal de Viçosa, Peter Henry Rolfs Avenue, Viçosa 36570–900, Brazil; lucianarenno@ufv.br (L.N.R.); samira.moreira@ufv.br (S.S.M.); hudsoncmartins@gmail.com (H.C.M.); isabela.iria@ufv.br (I.I.R.); detmann@ufv.br (E.D.); scvfilho@ufv.br (S.d.C.V.F.); mpaulino@ufv.br (M.F.P.); 2Eastern Oregon Agricultural Research Center, Department of Animal and Rangeland Sciences, Oregon State University, Burns, OR 97720, USA; matheus.lana@ufv.br

**Keywords:** cattle, behaviors, physiology, cortisol, number of vocalizations, ceruloplasmin, grazing, red blood cells

## Abstract

**Simple Summary:**

Although weaning is a necessary and indispensable practice for the beef cattle production system, it is known to cause multiple stressors in the cow–calf pair. However, few studies have explored the effects of stress on weaning in the dams, and little is known about the effect of parity on stress responses. The objective of this study was to investigate whether parity would influence the weaning stress response in beef cows under grazing. Our study shows that weaning impacted the behavioral and physiological parameters of Nellore cows under grazing, whereby multiparous cows displayed greater physiological changes and exhibited marked distress. Therefore, understanding how the stress response occurs during this period provides important information for better adaptation of management practices to increase animal welfare.

**Abstract:**

Most studies investigate the impact of stress at weaning on calves; however, little is known about the responses of cows, and whether they would differ according to parity. This study aims to investigate whether parity would influence the weaning stress response in beef cows. Thirty pregnant Nellore cows with their respective calves were randomly allocated to five paddocks and two females from each parity group were placed in the paddocks. There was an interaction (*p* < 0.05) between parity and evaluation days regarding cortisol, where on d + 7, the higher concentration was observed for multiparous cows. There was an interaction (*p* < 0.05) between parity and evaluation day for red blood cells (RBC), hematocrit (HCT), and hemoglobin (HB), whereby higher RBC counts on d + 4 were observed for multiparous cows. For HCT and HB, on all post-weaning collection days, higher values were observed for multiparous cows. The day of evaluation had an (*p* < 0.05) effect on all recorded behaviors, except for rumination (*p* > 0.05). Nellore cows, regardless of parity, underwent behavioral and physiological changes on abrupt weaning. Physiological parameters indicated that the magnitude of stress was greater in multiparous cows.

## 1. Introduction

Weaning is a necessary and indispensable management process that allows the shortening of the production cycle as it permits the separation of the calf–cow pair earlier than would occur naturally. However, weaning is an extremely sensitive period from the production point of view as it is known to cause multiple stressors in the cow–calf pair [1,2,3].

Weaning triggers several physiological responses in calves, such as increased serum concentrations of cortisol and acute phase proteins. These physiological changes, which are negatively associated with the performance and health of calves, have been the focus of the majority of studies in this field [4,5,6]. However, the weaning period can also have an impact on the behavior of cows. For example, cows have been shown to decrease grazing time by up to 50% and increase vocalization, pacing, and time spent looking for offspring after weaning [7,8,9,10]. Nonetheless, few investigations of physiological responses have been carried out in cows [11,12].

There are many factors that have been identified as influencing the stress response to abrupt weaning in beef cows, such as the individual characteristics of the mother and offspring, including the age and weight gain of the calf, breed, parity, physiological state of the cow, and milk production [3,9,12]. For early weaning management, parity has been shown to influence the behavioral stress response in beef cows, with multiparous cows experiencing marked distress when separated from their calves; however, the multiparous cows were all in their second calving (secundiparous cows), so the authors could not confirm whether, in cows with longer maternal experience (≥3 calvings), the differences would be similar or greater [12]. Multiparous cows exhibit better maternal behavior, as indicated by the fact that they nurse their calves more frequently [13] and produce more milk [14] than primiparous cows. The effects of parity on stress response at weaning seem to be related to the cow–calf bond, which is influenced by natural factors such as maternal ability and milk production. Therefore, the stronger the cow–calf bond, the more disruptive will be the weaning process.

Nevertheless, there is limited research that has examined weaning stress in the cow, and little is known about the extent of the effects of these physiological and behavioral changes. Further investigation is required to understand whether such responses are affected to the same extent when weaning is performed in the conventional way (at 7–8 months of age) and whether they are also influenced by parity. It is known that weaning stress affects the performance of beef cattle [15], and the metabolic status of the cow is influenced by parity, in which primiparous cows are more impacted by the lactation period, presenting more unbalanced metabolic and hormonal characteristics and lower body condition scores [16]. Therefore, understanding whether the response to stress in this period will have a negative impact on the productive life of the cow and whether it differs between categories is essential to adopting appropriate management practices aiming at greater animal welfare and productive and reproductive performance.

Since no study has evaluated whether physiological and behavioral responses differ between primiparous, secundiparous, and multiparous cows, we aimed to investigate the effect of parity on the weaning stress response in Nellore cows under grazing. Thus, we hypothesize that older cows will exhibit greater physiological and behavioral changes than young cows.

## 2. Materials and Methods

### 2.1. Experimental Design and Animal Ethics

All animal care and handling procedures were approved by the Animal Care and Use Committee of the Universidade Federal de Viçosa, Brazil (protocol CEUAP-UFV No. 071/2019). The experiment was conducted at the Department of Animal Science of the Universidade Federal de Viçosa, Viçosa, Minas Gerais, Brazil (20°45′S 42°52′ W), between June and July 2019. The experimental period of 34 d was divided into a pre-weaning phase (20 d) and a post-weaning phase (14 d). The average rainfall during the data collection period was 0.0000 mm, and the maximum and minimum average temperatures were 18 °C and 17 °C, respectively.

Thirty pregnant Nellore cows with their respective female Nellore calves were used in the study. The average ages of the 10 primiparous, 10 secundiparous, and 10 multiparous cows were 2 years, 3 years, and 4–6 years, respectively, while the average body weights (BW) were 451 ± 61 kg, 492 ± 43 kg, and 537 ± 43 kg, respectively. The cows used in this study calved between September and October 2018 and came from the same beef herd. Cow–calf pairs were randomly allocated to five paddocks covered with Uruchloa decumbens grass (8.7 ha/paddock), and two females from each parity group were placed in the paddocks (six cow–calf pairs/paddock) 28 days before the beginning of the experiment to acclimate to the environment and social interactions.

All animals had free access to water and a commercial mineral mix (CaHPO4, 50.00%; NaCl, 47.775%; ZnSO_4_, 1.4%; Cu_2_SO_4_, 0.70%; CoSO_4_, 0.05%; KIO_3_, 0.05%; and MnSO_4_, 0.025%). During the pre-weaning phase, calves were group-fed (six calves/paddock) with 6 g/kg BW of an energy-protein supplement containing 300 g crude protein (CP)/kg dry matter (DM) in a creep-feeding system. The supplement was formulated to supply approximately 50% of the CP requirement of calves, based on the BR-Corte system [17]. The supplement was provided daily at 1100 h. The average DM availability of forage for the experimental period was 5.0 t/ha (dry season). The chemical composition of forage by hand plucking was 380.7 g MS/kg of natural matter, 58.6 g CP/kg DM, 689.3 g neutral detergent fiber corrected for ash and protein/kg DM, 230.6 g indigestible neutral detergent fiber/kg DM, and 924.2 g organic matter/kg DM.

Calves were abruptly weaned (d 0) at an average age of 7.5 months. They were then loaded onto a livestock trailer and transported for 1.6 km to a different experimental area located at a sufficient distance away to avoid auditive interaction with the dams. The cows were kept in the same paddocks throughout the experiment, maintaining the social relationships established before weaning.

### 2.2. Sampling and Analysis

The cows were weighed (unshrunk BW) at d − 5 before cow–calf separation (pre-weaning baseline) and at d + 4, d + 7, and d + 14 (post-weaning) at 08:00 h. Simultaneously, blood samples were collected by jugular vein puncture using vacuum tubes with a clot activator and gel for serum separation (BD Vacutainer^®^ SST^®^ II Advance^®^, São Paulo, Brazil) to quantify cortisol, total proteins, and albumin. The BD Vacutainer^®^ Plus tube with sodium heparin was used for the quantification of haptoglobin and ceruloplasmin. After collection, samples were centrifuged at 2200× *g* for 20 min. Serum and plasma were frozen immediately and kept at −20 °C until they underwent analysis. Blood smears were prepared from whole blood with EDTA (BD Vacutainer^®^ EDTA, São Paulo, Brazil) for hematological analysis.

The behavior of individual cows was recorded on d − 3, +1, +3, and +6. Cows were tagged, and large numbers were painted on both sides of the body before observations were performed. The behavior was recorded by trained personnel (one per paddock), who were approximately 50 m away from the cows, using binoculars, to avoid the influence of human interference on the normal behavior of the cows [18,19]. Data were recorded from 06:00 h to 18:00 h by 15 evaluators, where each was responsible for one paddock for 4 consecutive hours. The observed behavioral characteristics were nursing time (on d − 3 only), grazing, ruminating, trough time, idle time (lying and standing), walking with the head held high (looking for the calf), pacing, and vocalization (Table 1).

The number of observations of each behavior was transformed into percentages, with the different behaviors evaluated through the daytime totaling 100% (grazing, ruminating, trough time, idle time, walking, and pacing); nursing time was added for pre-weaning assessment only. The frequency of vocalizations was recorded for each cow and expressed as the number of vocalizations per day (each call was recorded as one occurrence).

Serum cortisol concentrations were detected by the chemiluminescent method (Beckman Coulter^®^, 33600, Brea, CA, USA) using the Access^®^ 2 analyzer system (Beckman Coulter^®^, Brea, CA, USA). Blood concentrations of total protein (colorimetric kinetic test, Bioclin^®^, K031) and albumin (bromecresol green method, Bioclin^®^ K040, Belo Horizonte, Brazil) were quantified using an automated biochemical analyzer (Mindray, BS200E, Shenzhen, China).

Plasma ceruloplasmin was quantified by the oxidase activity method using colorimetric procedures, as described by Demetriou et al. [21]. Plasma haptoglobin concentration was quantified in duplicate samples by estimation of differences in peroxidase activity caused by haptoglobin–hemoglobin complexing [22]; the results were expressed as optical density resulting from readings at 450 nm × 100 [23].

Red blood cell (RBC) and white blood cell (WBC) count and hemoglobin concentration (HB) were quantified using the Hematoclin 2.8 Vet automatic analyzer (Bioclin^®^, Belo Horizonte, Brazil), the differentiation of leukocyte cells (neutrophils and lymphocytes) was determined under a microscope, and neutrophil to lymphocyte (N:L) ratios were calculated. Hematocrit (HCT) was quantified in a microcentrifuge (MicroSpin microhematocrit centrifuge, Jaboticabal, Brazil).

### 2.3. Statistical Analysis

The experiment was analyzed using the model:
$$Y_{ijk}=\mu +P_{i}+O_{j}+e_{(ij)k}$$
where *Y_ijk_* is the observation taken on animal *k*, pertaining to parity *j*, and managed in the paddock i; μ is the overall constant; *P_i_* is the random effect of paddock *i; O_j_* is the fixed effect of parity *j*; and *e_(ij)_*_k_ is the random error, assumed to be NIID (*0, σ2 e*).

All response variables were interpreted as repeated measurements [24] according to the evaluation days around the weaning time (fixed effect). The best structure of the residual (co)variance matrix was chosen based on Akaike’s information criterion with correction. Degrees of freedom were estimated using the Kenward–Roger method. The analyses were performed using PROC MIXED in SAS 9.4 (Inst. Inc., Cary, NC, USA), with significance set at *p* < 0.05.

After a first round of analysis of variance, the experimental error was estimated and evaluated with regard to normal distribution and homoscedasticity (across parity and evaluation day) using the approaches provided by the Shapiro–Wilk and Levene tests, respectively. When these criteria were matched, pairwise comparisons were performed using the protected Fisher’s least significant difference test.

We anticipated that experimental errors for vocalization would present neither normal distribution nor homogeneous variance (*p* < 0.05). Attempts were made to use the basic experimental model with different probability distributions for the errors through the GLIMMIX procedure of SAS 9.4. However, adequate convergence was not achieved. Hence, the pattern of vocalizations across evaluation days and parity was evaluated using a non-parametric Kruskal–Wallis (KW) approach via the NPAR1WAY procedure in SAS 9.4. When an overall significance for the KW test was achieved (*p* < 0.05), the Wilcoxon scores test was applied as a pairwise comparison procedure.

## 3. Results

### 3.1. Body Weight

Analysis of the cows’ BW showed that there was no interaction (*p* = 0.343) between parity and evaluation day (Figure 1). On average, multiparous cows presented a higher BW (*p* < 0.0001; SEM [Standard error of means] = 17.45), followed by secundiparous and primiparous cows. The BW of the cows changed according to the day of evaluation (*p* < 0.0001), with the lowest average BW observed just after the weaning.

### 3.2. Physiological Measurements

Analysis of serum cortisol concentrations in the blood revealed an interaction (*p* = 0.038; SEM = 1.03) between parity and evaluation day (Table 2). Furthermore, differences between parity orders were detected (Figure 2a) on d + 7, with higher cortisol concentrations detected in multiparous cows than in the other parity orders, which were similar to each other (*p* > 0.05). However, effects of day, but not parity or parity and day, were detected for ceruloplasmin (*p* < 0.0001; SEM = 0.99; Table 2) and haptoglobin concentrations (*p* < 0.0001; SEM = 0.58; Table 2). In addition, ceruloplasmin (Figure 2b) and haptoglobin (Figure 2c) concentrations increased post-weaning and did not return to pre-weaning values on d + 14.

Total protein (*p* = 0.0003; SEM = 0.17) and albumin (*p* = 0.049; SEM = 0.08) concentrations were affected by parity, with different values between the parity orders already in the pre-weaning phase (Table 2). Additionally, total protein (*p* = 0.0002) and albumin (*p* = 0.0001) concentrations were affected by day. Total protein concentrations were higher for multiparous and secundiparous cows than for primiparous cows, with the lowest concentrations detected on d + 14 (Figure 3a). Furthermore, albumin levels were higher for multiparous cows than for primiparous cows and decreased from d + 7, with lower concentrations on d + 14 (Figure 3b).

For RBC count, there was an interaction (*p* = 0.014; 0.27) between parity and evaluation day (Table 2). Differences between parity orders were detected on d + 4, whereby higher RBC counts were observed for multiparous cows (Figure 4a) compared with secundiparous and primiparous cows, which did not differ from each other. An interaction was also observed (Table 2) between parity and evaluation days for HCT percentage (*p* = 0.009; SEM = 0.75). Differences between parity orders were detected on all post-weaning collection days (Figure 4b), with higher values being observed for multiparous cows than for primiparous cows. In addition, on d + 4 and d + 14, HCT values were higher for multiparous cows than for secundiparous cows. However, HCT values were only higher for secundiparous cows compared to primiparous cows on d + 7. Analysis of HB concentration also revealed an interaction (*p* = 0.034; SEM = 0.29; Table 2) between parity and evaluation day. On all post-weaning collection days, HB concentration (Figure 4c) was higher for multiparous cows than for primiparous cows.

In addition, effects of day relative to weaning were detected for WBC counts and lymphocyte and neutrophil numbers (Table 2). All cows had lower WBC counts after weaning (*p* < 0.0001; SEM = 0.76; Figure 5a). Lymphocyte numbers decreased on d + 4 and d + 7 and increased again on d + 14 (*p* = 0.004; SEM = 0.54; Figure 5b). Neutrophil numbers were also lower on d + 4 (*p* = 0.006; SEM = 0.23; Figure 5c). Nonetheless, the N:L ratio was not affected by parity, day, or parity and day (*p* > 0.05; Table 2).

### 3.3. Behavior Characteristics

There were no differences in time spent nursing before weaning (*p* > 0.05; Table 3). In addition, behavioral observations indicated that there was no effect of parity or interaction between parity and evaluation day (*p* > 0.05; Table 3). However, the day of evaluation had a significant (*p* < 0.05) effect on all recorded behaviors, except for rumination (*p* = 0.625; SEM = 1.42). Time spent grazing decreased on d + 1 after weaning, increasing on d + 3 and d + 6 but remaining lower than the pre-weaning values (*p* < 0.0001; SEM = 2.85; Figure 6a). Idle time increased after weaning (*p* = 0.013; SEM = 1.97; Figure 6b), while cows displayed lower trough time on the first day after weaning, returning to pre-weaning time on d + 6 (*p* = 0.037; SEM = 0.35; Figure 6c). Pacing (*p* < 0.0001; SEM = 1.55; Figure 6d) and walking (*p* < 0.0001; SEM = 0.59; Figure 6e) were higher on d + 1 after weaning than on the other days.

Evaluation day also had an effect on the number of vocalizations by the cows (Table 3). On average, the cows vocalized more on the first day after separation from the calf (*p* < 0.0001; Figure 7a). As this specific day was the only one to present a significant number of vocalizations, we performed a nested evaluation to understand the differences between parity orders within the first day after weaning. However, despite a numerical trend whereby the older cows produced a greater number of vocalizations, no significant differences between parity orders were detected (*p* = 0.580; SEM = 17.86; Figure 7b).

## 4. Discussion

Weaning is a dynamic process, the underlying effects of which can impact different pathways. As such, assessing only changes in physiology, behavior, or performance provides limited information [25]. Therefore, this study thoroughly evaluated weaning stress responses in beef cows using multiple approaches, enabling a broader view of the effects of weaning.

We showed that breaking the mother–offspring bond under conventional weaning management (weaning at 7 to 8 months) represented a stressful experience for Nellore cows, causing physiological and behavioral changes. Those changes were mostly observed during the early post-weaning period. Similarly, other authors have found behavioral responses to the stress of abrupt weaning to be short-lived, showing that most behaviors returned or were returning to pre-weaning frequencies by the third day after weaning [8,12]. Indeed, Lenner et al. [26] demonstrated that when cow and calf were brought together in the first week after weaning, the bond was stronger than when they were brought together during the third week.

Furthermore, in our study, although the behavior did seem not to be influenced by parity, there was physiological evidence that the magnitude of stress was greater for multiparous cows. Stěhulová et al. [9] showed that parity had no effect on the behavior of pregnant beef cows when weaning was performed at 7 months of age. The authors pointed out that reactions were reduced in pregnant cows or cows with older calves. As pregnant cows redirect resources to future offspring [27,28], the magnitude of behavioral changes is masked by these effects. Nonetheless, one of the first studies to evaluate stress responses to early weaning in beef cows of different parity orders showed that stress responses were stronger in multiparous cows, as indicated by greater changes in behavior such as walking and vocalizing. The authors concluded that weaning affected the welfare of multiparous cows more than that of primiparous cows, but these outcomes did not extend to physiological parameters [12]. It is noteworthy that in the experiment by Ungerfeld et al. [12], assessments were performed on multiparous cows that were all on their second calving, which corresponds to our secundiparous cows, and it is important to emphasize that secundiparous cows present intermediate characteristics between primiparous and multiparous. The differences between studies evaluating behavioral and physiological changes during abrupt weaning could be related to the different weaning ages, as in early weaning, the stress is more remarkable because of the stronger cow–calf bond [29]. In addition, they may be related to breeding, since cow temperament appears to influence the maternal ability [30] and, consequently, the stress of separation [3,31].

Differences in BW found between parity orders were expected because of age differences [32]. However, all categories showed reduced BW after weaning, which is indicative of stress caused by weaning. As the cows showed reduced trough and grazing time, the weight changes may have been related to lower water and feed intake, and thus less filling of the gastrointestinal tract [33]. In addition, the cows increased the amount of idle time, as well as time spent walking with their heads raised and time looking for their calves parallel to the fence. Cows have been reported to walk more when permanently separated from their calves [2] and to remain close to fences after weaning [8]. Thus, cows appear to spend more time looking for their calves than grazing, which may be intensified depending on the magnitude of the weaning stress [34]. This nutrient deprivation may stimulate the mobilization of body fat reserves and activate the hypothalamic–pituitary–adrenal axis [35,36], triggering an acute phase response in cattle [37,38]. Water and feed deprivation can also interfere with the ruminal environment, causing microbial death [39] and resulting in the release of endotoxins that can induce an acute phase response [4]. Thus, reduced dry matter intake has been identified as a contributor to acute phase responses and reduced performance in beef cattle [33,40,41,42].

Cortisol is indicated as a modulator of stress adaptation. One of the physiological effects of cortisol during the stress response is the triggering of nutrient mobilization from the liver, fat, and muscle [43]. This catabolic effect increases the amount of circulating nutrients available for the animal to deal with the stressor and restore homeostasis. However, defense cells interpret this tissue mobilization as a deviation from homeostasis and initiate the production of cytokines, which are the main stimulators of acute phase protein production in the liver [44]. The main positive acute phase proteins that increase during inflammation or stress in cattle are haptoglobin, ceruloplasmin, and amyloid A [42,45,46,47]. Our results showed that ceruloplasmin and haptoglobin concentrations increased after weaning in all cows, regardless of parity, although increased cortisol was only detected in multiparous cows. Similarly, Lynch et al. [11] showed that haptoglobin concentration was increased in primiparous cows after weaning, without changes in cortisol concentration.

Differences found in total proteins and albumin concentrations according to parity are not related to weaning stress. According to Ferreira et al. [16], parity directly influences all indicators of protein status (i.e., total proteins, albumin, globulins, urea, and IGF-1) in beef cows. Thus, protein concentrations are expected to be lower for primiparous cows, as this category requires nutrients for fetal development, lactation, and continued growth. Nonetheless, regardless of category, there are two possible reasons that may explain the reduction in albumin concentration on days +7 and +14 after weaning. As previously described, there was a reduction in grazing, implying that protein intake was decreased, which could have led to lower serum albumin concentrations [48]. At the same time, this reduction may have been potentiated by the stress of weaning, as albumin is also known as a negative acute phase protein [46]. Determination of acute phase protein levels is used to monitor the health and welfare of animals in large herds [47]. Thus, low albumin concentration and high levels of haptoglobin and ceruloplasmin can be indicative of distress in cows when separated from their calves.

The increases in RBC count, HB concentration, and HCT percentage for multiparous cows after weaning did not appear to be related to polycythemia. Instead, these altered parameters may have been a reflection of lower water intake, although trough time was similarly reduced in the different parity orders. Moreover, it should be noted that multiparous cows had a cortisol peak, suggesting that this increase serves to protect and maintain water balance in times of stress [49]. Similarly, Campistol et al. [6] reported increases in RBCs and HCT in calves after weaning. Additionally, alterations in WBC population components occurred regardless of parity, indicating an attempt to restore homeostasis. WBC numbers were reduced, with lower numbers of lymphocytes (the main WBC type in cattle), neutrophils, and monocytes (data not shown). According to Lynch et al. [11], in beef cows, leukocyte and lymphocyte numbers decrease after weaning, which is characteristic of a stress leukogram. However, despite the post-weaning reductions, the RBC and WBC counts reported for all cows were in line with the reference limits for beef cows [50].

Previous studies from our group found that parity influenced milk production and composition in beef cows. Milk yields were higher in multiparous animals than in primiparous and secundiparous cows [14], in line with other studies [51,52]. Maternal milk output represents the costliest investment of the cow in the growth of her calf. Thus, mothers of calves with greater weight gain would experience greater marked suffering because milk is the most important factor in the weight gain of their calves. Grazing time has also been observed to be higher for calves born to primiparous and secundiparous cows than for those born to older animals [53]. Calves born to cows with lower milk production tend to increase grazing in an attempt to consume similar amounts of metabolizable energy per unit weight [54,55]. Conversely, calves of multiparous cows are expected to receive more milk as a result of the higher milk production of the dams; therefore, the cow–calf bond should be greater in such pairs. It is interesting to note that our findings were consistent with these results, indicating that female calves born to one- or two-calf cows exhibit more independence and thus, weaker cow–calf bonds. This may have contributed to the lower stress responses in these cows during weaning. Nonetheless, a previous study indicated that weight gain in calves was less important than the age of the calf at weaning in terms of effects on the behavior of cows [9].

In summary, our study is the first to show that in Nellore cows, behavioral stress responses to abrupt weaning were independent of previous maternal experiences and were of short duration. They were characterized mainly by the initiation of pacing, walking, and vocalization, and reduced grazing and trough time on the first day after cow–calf separation. In contrast, physiological changes lasted longer and were more pronounced in multiparous cows who displayed higher cortisol levels, erythrocyte numbers, HB concentrations, and HCT percentages. With these findings, specific practices that aim to improve the well-being and reproductive performance of cows, especially multiparous cows, can be developed and adapted to avoid the mistaken use of management that was generated from information on other breeds, other types of weaning, and environments. More research on this topic should be developed to establish specific practical recommendations for these animals, such as the use of alternative methods to abrupt weaning for multiparous cows.

## 5. Conclusions

Nellore cows, regardless of parity, underwent behavioral and physiological changes as a result of abrupt weaning. Additionally, physiological parameters indicated that the magnitude of stress was greater in multiparous cows.

## Figures and Tables

**Figure 1 animals-13-01321-f001:**
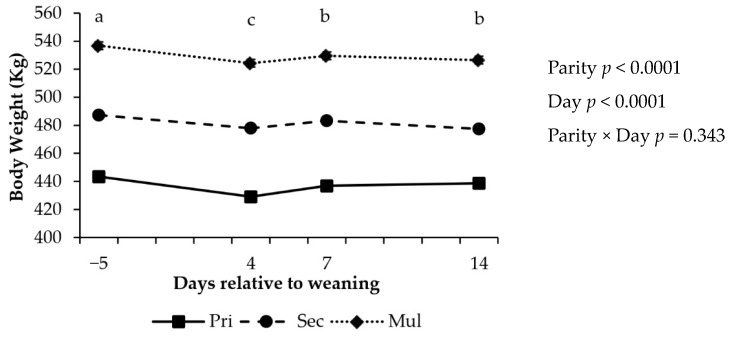
The body weight of Nellore cows with different parity under grazing according to the days relative to weaning. Days with different superscripts differ from each other for all parity orders (*p* < 0.05).

**Figure 2 animals-13-01321-f002:**
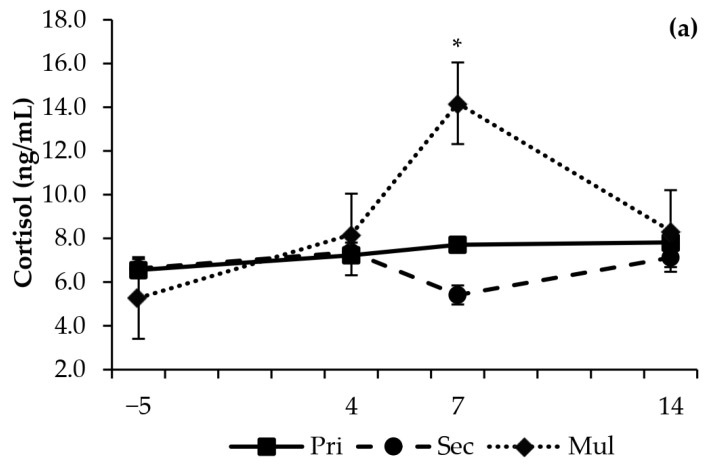
Cortisol (**a**), ceruloplasmin (**b**), and haptoglobin (**c**) serum concentrations in Nellore cows with different parity under grazing according to the days relative to weaning. Days with different superscripts differ from each other for all parity orders (*p* < 0.05). On days with asterisks (*), there is a parity × day interaction (*p* < 0.05).

**Figure 3 animals-13-01321-f003:**
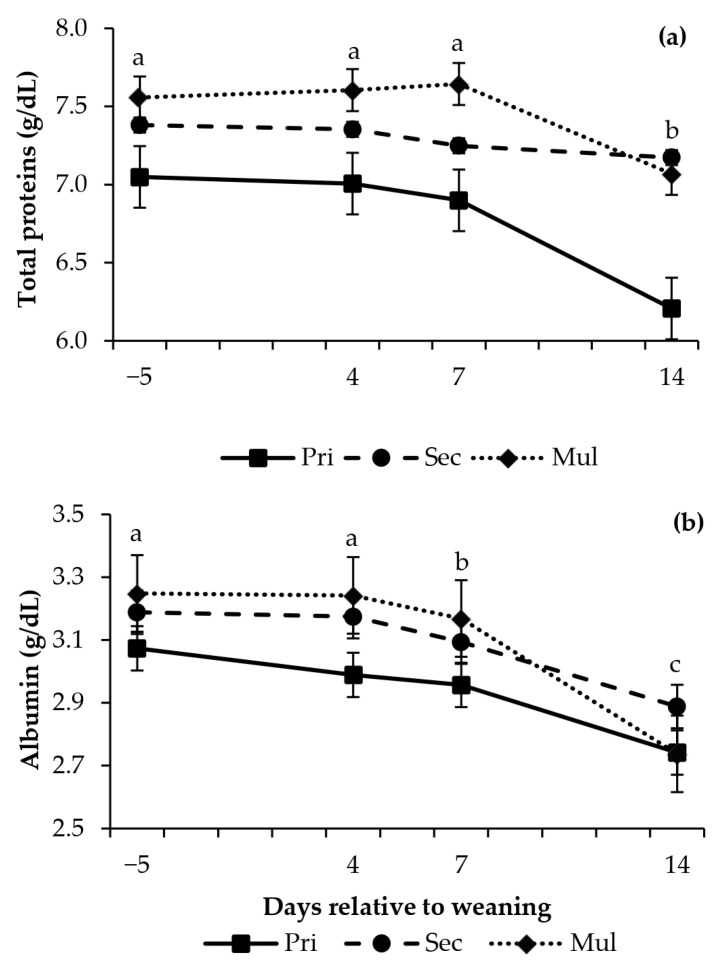
Total proteins (**a**) and albumin (**b**) serum concentrations in Nellore cows with different parity under grazing according to the days relative to weaning. Days with different superscripts differ from each other for all parity orders (*p* < 0.05).

**Figure 4 animals-13-01321-f004:**
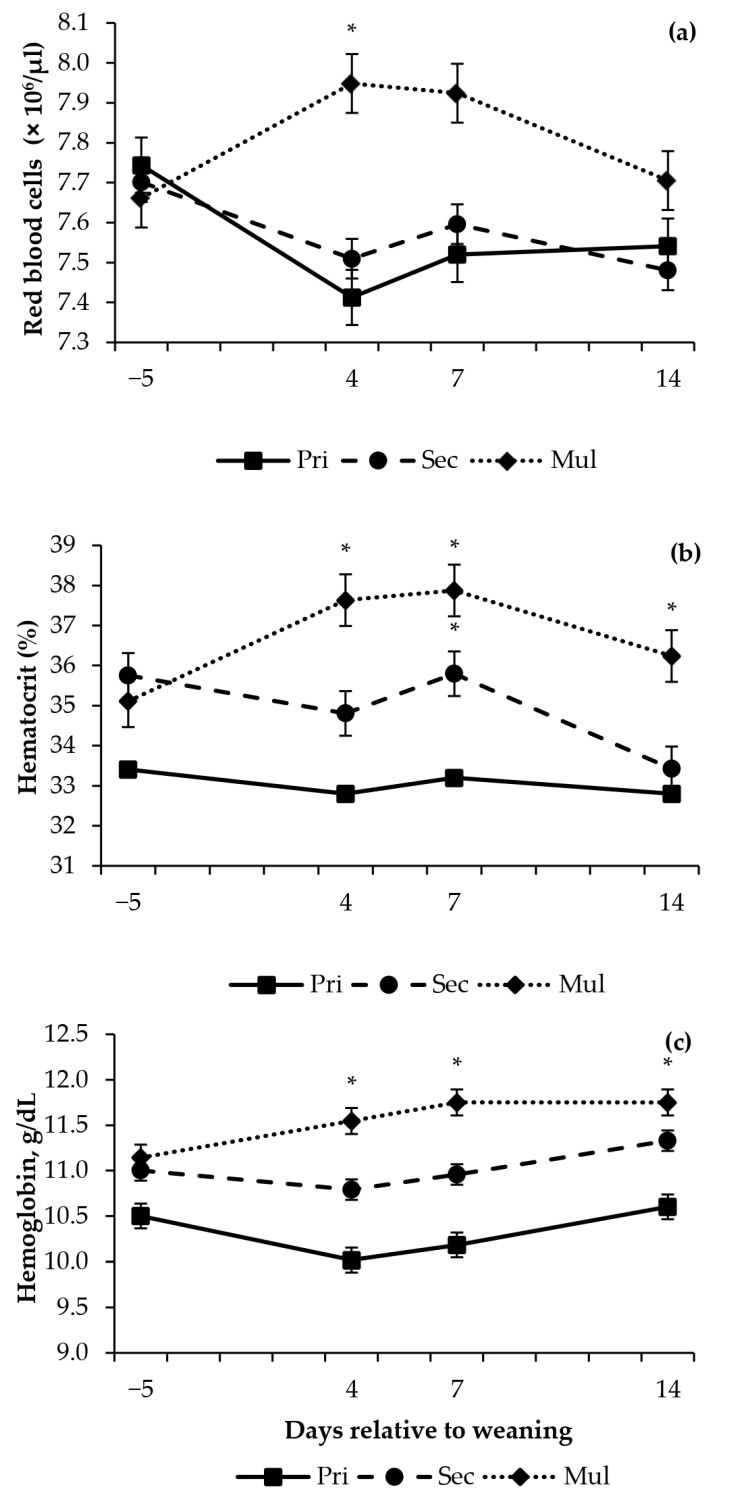
Red blood cell number (RBC; (**a**)), Hematocrit percentage (HCT; (**b**)), and hemoglobin concentration (HB; (**c**)) in Nellore cows with different parity under grazing according to the days relative to weaning. On days with asterisks (*) there is a parity x day interaction (*p* < 0.05).

**Figure 5 animals-13-01321-f005:**
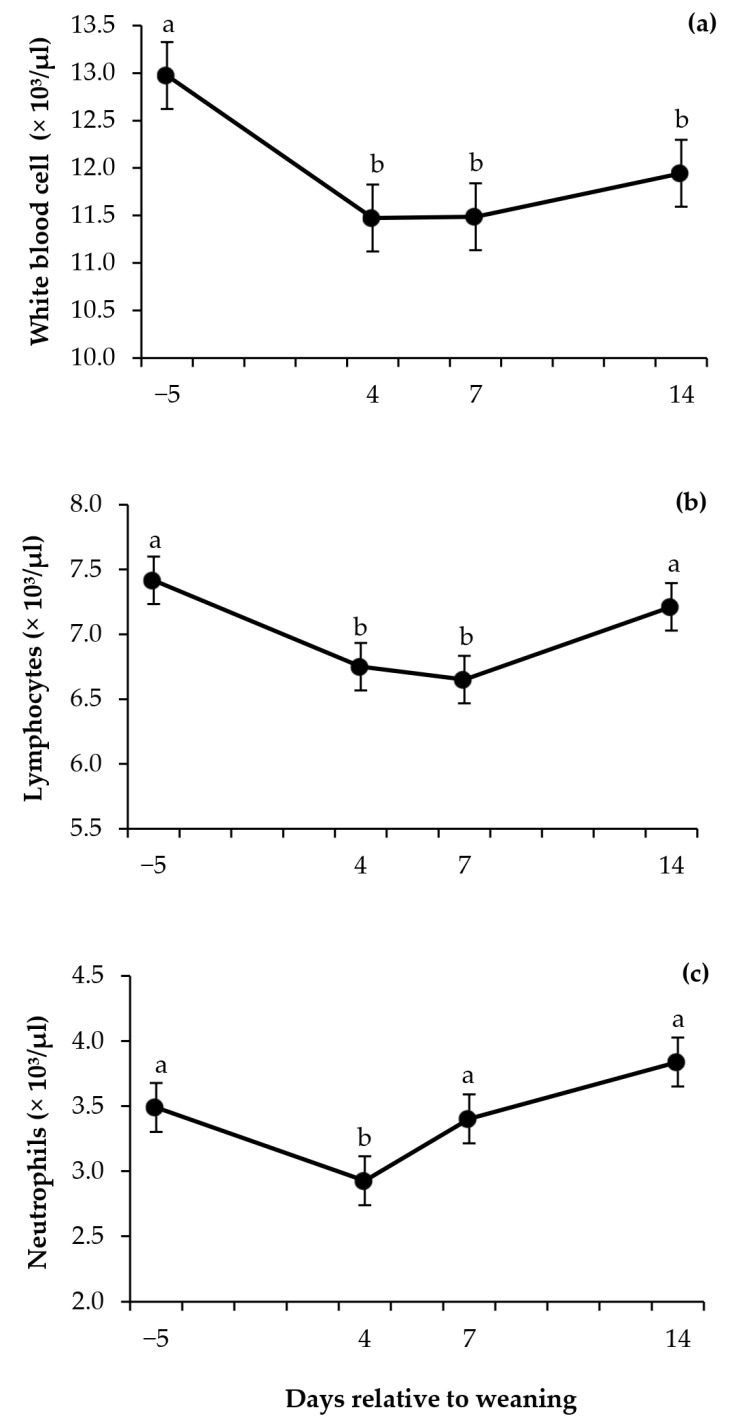
White blood cell number (WBC; (**a**)), lymphocyte number (**b**), and neutrophil number (**c**) in Nellore cows with different parity under grazing according to the days relative to weaning. Days with different superscripts differ from each other for all parity orders (*p* < 0.05).

**Figure 6 animals-13-01321-f006:**
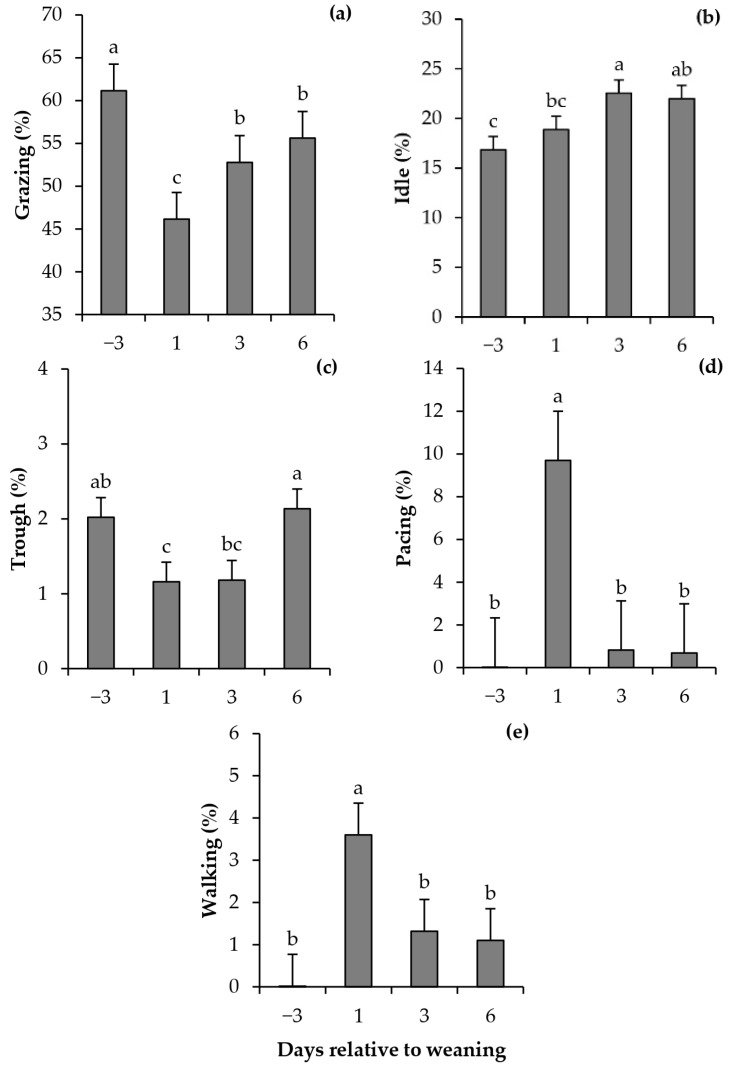
Grazing (**a**), idle time (**b**), trough time (**c**), pacing (**d**), and walking (**e**) in Nellore cows under grazing according to the days relative to weaning. Days with different superscripts differ from each other (*p* < 0.05).

**Figure 7 animals-13-01321-f007:**
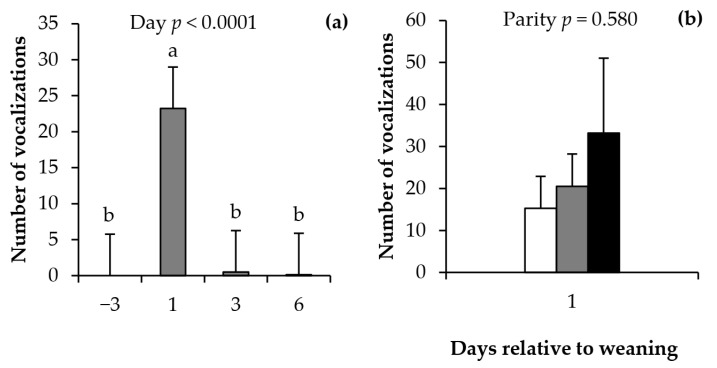
The number of vocalizations recorded in Nellore cows under grazing according to the days relative to weaning (**a**) and in primiparous (white), secundiparous (grey), and multiparous cows (black) on d + 1 after weaning (**b**). Days with different superscripts differ from each other (*p* < 0.05).

**Table 1 animals-13-01321-t001:** Description of observed cattle behaviors in pasture.

Behavior	Definition *
Grazing (% ^1^)	Picking or consuming pasture, with the head above ground
Idle (%)	In any resting position
Ruminating (%)	Chewing regurgitated boluses of feed
Trough (%)	Going to the trough in search of food or water
Pacing (%)	Moves looking for the calf parallel to the fence, at 1 m
Walking (%)	All four legs are moved with the head raised
Nursing (%)	The calf suckles the udder
Vocalization ^2^	Emission of characteristic sounds through the mouth

* Adapted from [20]. ^1^ % = percentage of observation time; ^2^ Number of vocalizations.

**Table 2 animals-13-01321-t002:** Effect of parity on physiological measurements in Nellore cows at abrupt weaning.

	Parity				*p*-Value		
Item	Primiparous	Secundiparous	Multiparous	SEM	Par	Day ^1^	Par × Day
Cortisol, ng/mL	7.33	6.63	9.00	1.031	0.198	0.174	0.038
Ceruloplasmin, mg/dL	12.60	11.52	10.36	0.986	0.157	<0.0001	0.372
Haptoglobin, DO×100	3.33	3.44	3.23	0.579	0.578	<0.0001	0.081
Total proteins, g/dL	6.79 ^b^	7.29 ^a^	7.45 ^a^	0.165	0.0003	0.0002	0.196
Albumin, g/dL	2.94 ^b^	3.09 ^ab^	3.10 ^a^	0.079	0.049	0.0001	0.668
RBC, ×10^6^/μL	7.55	7.57	7.81	0.207	0.337	0.287	0.014
HCT, %	33.05	34.94	36.71	0.748	0.001	0.011	0.009
HB, g/dL	10.33	11.02	11.55	0.292	0.001	0.002	0.034
WBC, ×10³/μL	12.25	11.79	11.86	0.756	0.866	<0.0001	0.390
Lymphocytes, ×10³/μL	7.07	7.03	6.92	0.538	0.974	0.004	0.947
Neutrophils, ×10³/μL	3.53	3.32	3.39	0.234	0.777	0.006	0.375
N:L	0.51	0.48	0.53	0.033	0.542	0.201	0.947

OD = Optical Density; RBC = Red Blood Cell number; HCT = Hematocrit percentage; HB = Hemoglobin concentration; WBC = White Blood Cell number; N:L = Neutrophil/Lymphocyte relationship; SEM = Standard error of means; Par = Parity. ^1^ Day relative to weaning. ^a–b^ Different letters declare significantly different between parities at *p* < 0.05.

**Table 3 animals-13-01321-t003:** Effect of parity on behavior observation in Nellore cows at abrupt weaning.

	Parity				*p*-Value		
Item	Primiparous	Secundiparous	Multiparous	SEM	Par	Day ^1^	Par × Day
Grazing, %	55.71	53.41	52.62	2.853	0.515	<0.0001	0.613
Idle, %	17.95	20.41	21.82	1.973	0.259	0.013	0.959
Ruminating, %	20.34	19.34	19.17	1.418	0.769	0.625	0.411
Trough, %	1.47	1.64	1.75	0.348	0.729	0.037	0.926
Pacing, %	2.97	3.21	2.22	1.551	0.847	<0.0001	0.995
Walking, %	1.10	1.78	1.64	0.587	0.091	<0.0001	0.798
Nursing ^2^, %	1.58	1.58	1.72	0.503	0.962	-	-
Number of vocalizations	15.27	20.5	33.18	17.860	0.580		

SEM = Standard error of means; Par = Parity. ^1^ Day relative to weaning. ^2^ nursing time was added for pre-weaning assessment only.

## Data Availability

The data were not deposited in an official repository. The data generated during the current study are available from the corresponding author upon reasonable request.

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
