# Peer review of "Does Parity Influence the Magnitude of the Stress Response of Nellore Cows at Weaning?"

_animals, 2023, doi:10.3390/ani13081321_

Round 1
Reviewer 1 Report
Authors conducted a well-designed, well-written and interesting study regarding stress response in Nellore cows at weaning. I would like to congratulate authors because it is correctly designed and the statistics applied are right. I would encourage them to improve the introduction to show need of this study. As well, I have some other minor comments as follows.
Some other comments:
- ln 55-63. Some studies regarding parity order have been performed. Please show the need of this study.
- ln 83. why just female calves?
- ln 122. 50 min?
- figure 1. is it really a body weight loss?? I would delete "loss"
- figure 1. superscripts are just for primiparous?
Author Response
Response to Reviewer 1 Comments
Comments and Suggestions for Authors
Authors conducted a well-designed, well-written and interesting study regarding stress response in Nellore cows at weaning. I would like to congratulate authors because it is correctly designed, and the statistics applied are right. I would encourage them to improve the introduction to show need of this study. As well, I have some other minor comments as follows.
Author: Author: Thank you for your considerations. We are open to any changes and we’re happy to have your contributions to make our research better.
Some other comments:
- ln 55-63. Some studies regarding parity order have been performed. Please show the need of this study.
Author: Thank you for your suggestion. We've improved the introduction. We hope it was adequate, but we are open to further suggestions if necessary.
- ln 83. why just female calves?
Author: The animals used in this experiment are supplied by the beef cattle ranch of the Department of Animal Science of the Universidade Federal de Viçosa which performs inseminates with sexed semen, so we only had available cows with female calves.
- ln 122. 50 min?
Author: Thank you for your question. We thought the sentence was confusing, so we rewrote it. In truth, the observers stayed at a minimum distance of 50 m from the animals.
- figure 1. is it really a body weight loss?? I would delete "loss."
Author: Thank you for your question. Yes, you're right. I delete it.
- figure 1. superscripts are just for primiparous?
Author: Thank you for your question. No, superscripts are for all parity orders. I changed the way of placing the superscripts and I hope it turned out better, please see if it's ok.
Reviewer 2 Report
Overall, this article is written very well, the statistical analysis is thorough, and the biological mechanisms are clearly discussed.
Instead of writing “P<0.05” throughout the manuscript, I recommend putting in the effect size with standard errors. Since you already stated that statistical significance was set at P<0.05, it is not needed. Having the effect sizes in text will help reader understand magnitude of differences at a glance.
Was there a sample size calculation performed?
Did you consider adjusting for multiple testing?
Were any of these variables skewed and did you need to transform? Are these backtransformed values? From my experience, some of these markers tend to be skewed.
What are the big picture ramifications of this research? How will this affect management of cows, if it will? Discuss in intro and discussion.
L209-110: The ones with significant interactions were also affected by parity since they showed different trends based on parity.
Just a minor comment/suggestion. “Parity order” is confusing wording to me. You could just shorten to parity and use “parity group” throughout the manuscript.
L52-62: in these prior studies, when did weaning take place and how does that differ from this current study? Pardo my ignorance on the topic.
L69-71: Add citation.
Table 2 and Table 3: Report SEM for each group so the reader can see the variation between the groups (even if heteroskedasticity was not significant).
Graphs: Maybe in footnotes mark which ones did not have significant interaction (and that’s why there is only one line) because it looked confusing at first glance.
Figure 4: The last sentence, is each parity significant from each other? Which pairwise comparisons?
L296-299: In the discussion it states that “assessing only changes in physiology behavior, or performance provides limited information. Therefore, this study…” Do you mean that assessing changes in these measures separately provides limited information? Clarify here.
L312-313: Interpreting non-significant results. I would remove.
Have there been any studies in dairy cows to add to introduction?
L81: add digits or truly 0.0000?
L121: Behavior was recorded live via binoculars one per paddock for 12 h? Seems like there could be some room for error and should discuss this possibility in the discussion along with any other limitations.
What are the next steps of this research? Add to discussion/conclusion.
Figure 1: missing error bars
What covariate structures were considered?
Author Response
Response to Reviewer 2 Comments
Author: Thank you for your suggestions and corrections in our study. We are open to any changes and we’re happy to have your contributions to make our research better.
Reviewer 2: Instead of writing “P<0.05” throughout the manuscript, I recommend putting in the effect size with standard errors. Since you already stated that statistical significance was set at P<0.05, it is not needed. Having the effect sizes in text will help reader understand magnitude of differences at a glance.
Author: Thank you for your suggestions. You're right. I changed it.
Reviewer 2: Was there a sample size calculation performed?
Author: Thank you for your question. Sample size calculation was not performed. We used all our resources in this research to keep the number of cows balanced between treatments (10 primiparous, 10 secundiparous, 10 multiparous). We know that this calculation is necessary, however, we do not believe that the number of animals limited the results of our study. Prior to the experiment, we did a thorough investigation of the studies that evaluated weaning stress responses (behavioral and physiological responses), and we saw that they used a similar number of animals per treatment (Ungerfeld et al., 2009; Lynch et al., 2010; Lynch et al., 2011; Pérez-Torres et al., 2016; StÄ›hulová et al., 2017).
Reviewer 2: Did you consider adjusting for multiple testing?
Author: Thank you for your question. It was not performed directly. There were only three treatments being compared, which is little considering a possible "family-wise error rate”. On the other hand, we made all comparisons between treatments with the protection of the F-test (ANOVA), in the other words, the multiple comparison tests were only applied if the F-test showed significance (P < 0.05).
Reviewer 2: Were any of these variables skewed and did you need to transform? Are these back transformed values? From my experience, some of these markers tend to be skewed.
Author: Thank you for your question. In none of the variables was it necessary to transform. Only one of the evaluated variables (Number of vocalizations) did not accord with the criteria of normality and asymmetry. However, the number of vocalizations was assessed using a non-parametric test, thus not requiring transformation.
Reviewer 2: What are the big picture ramifications of this research? How will this affect management of cows, if it will? Discuss in intro and discussion.
Author: Thank you for your question and suggestion. The ramifications of the general picture of this research and the great differential are the characterization of weaning stress in Nellore cows, with physiological and behavioral responses together. Through this, we identified that multiparous cows have physiological responses that may reflect in lower productive performance, however, this needs further research. But we know that the weaning method must be adapted so that these cows feel less of the effects of abrupt weaning. We add to the intro and discussion.
Reviewer 2: L209-110: The ones with significant interactions were also affected by parity since they showed different trends based on parity.
Author: Thanks for your comment. Yes, you're right. I removed it.
Reviewer 2: Just a minor comment/suggestion. “Parity order” is confusing wording to me. You could just shorten to parity and use “parity group” throughout the manuscript.
Author: Thanks for your suggestion. I removed the word “order”.
Reviewer 2: L52-62: in these prior studies, when did weaning take place and how does that differ from this current study? Pardo my ignorance on the topic.
Author: Thanks for your question. Different weaning ages and their implications for the stress response have been studied. Some authors have reported that in early weaning (up to 90-120 days) stress would be more marked because the calf at this age is more dependent on the mother's care and milk. So, the intensity/magnitude of the responses declines with time (Lynch et al., 2019). Therefore, we added this information, because in this specific study that we put in the introduction, weaning was done 71 days postpartum (Ungerfeld et al., 2011). Thus, this information becomes relevant, as there is no study that jointly evaluated age at weaning and the effect of parity.
Reviewer 2: L69-71: Add citation.
Author: Thanks for your correction. We've reworked the intro and that sentence has been modified.
Reviewer 2: Table 2 and Table 3: Report SEM for each group so the reader can see the variation between the groups (even if heteroskedasticity was not significant).
Author: Thank you for your suggestion. The variables presented in tables 2 and 3 showed homoscedasticity. Thus, we chose to expose a single SEM, which refers to the variability of the general mean for each treatment and corresponds to the solution provided by PROC MIXED of SAS. We opted for this, as we understood that it would make the tables cleaner to facilitate the reader's understanding. However, if you still think it's necessary, we can add the SEM of each group.
Reviewer 2: Graphs: Maybe in footnotes mark which ones did not have significant interaction (and that’s why there is only one line) because it looked confusing at first glance.
Author: Thank you for your suggestion. I changed de footnotes, please see if it's ok.
Reviewer 2: Figure 4: The last sentence, is each parity significant from each other? Which pairwise comparisons?
Author: Thank you for your question. I changed it.
Reviewer 2: L296-299: In the discussion it states that “assessing only changes in physiology behavior, or performance provides limited information. Therefore, this study…” Do you mean that assessing changes in these measures separately provides limited information? Clarify here.
Author: Thanks for your question. Yes, assessing only physiological or behavioral variables will show only part of the weaning stress response. Because the behavioral stress response is short-lived, as around 3 days after weaning, most behaviors return to baseline frequencies, but physiological changes last longer (two weeks). In addition, there is still controversy as to which are the best markers of the weaning stress response, since in some studies there are only behavioral changes (Ungerfeld et al., 2011) and in others, there is a more significant physiological response. Therefore, to understand the response and whether it will affect animal welfare and performance, multiple approaches will draw a more complete profile. Unfortunately, joint studies with physiological and behavioral assessments are rare, probably due to the difficulty of data collection, and that is why our study is so important to know how occurs the response to weaning stress in beef cattle.
Reviewer 2: L312-313: Interpreting non-significant results. I would remove.
Author: Thanks for your suggestion. I removed it.
Reviewer 2: Have there been any studies in dairy cows to add to introduction?
Author: Thanks for your question. There are studies on dairy cows, however, we chose not to use these references because the handling of dairy cattle is different from beef cattle. For example, in dairy cows, the calf is weaned much earlier, in many cases, the calf only has contact with its dam in the first days of life, and the form of milk supply is artificial. Thus, we believe that mother–offspring bond is less intense, and it could cause mistaken conclusions, so would not contribute to our study.
Reviewer 2: L81: add digits or truly 0.0000?
Author: Thanks for your suggestion. Yes, you're right. I added it.
Reviewer 2: L121: Behavior was recorded live via binoculars one per paddock for 12 h? Seems like there could be some room for error and should discuss this possibility in the discussion along with any other limitations.
Author: Thanks for your question. Yes, the behavior was recorded live via binoculars (if necessary). However, the observers were changed every 4 hours of observation, thus totaling three evaluators per paddock. We do not believe this is a limitation, as the observers were trained, and this methodology was based on previous studies (Valente et al., 2013; Lopes et al., 2017; Martins et al., 2017). To clarify, we have added this information to the materials and methods.
Reviewer 2: What are the next steps of this research? Add to discussion/conclusion.
Author: Thank you for your question and suggestion. The next steps of this research are studies that aim for alternative weaning methods, mainly for multiparous cows, and new research to analyze the impacts of this stress on the well-being and future performance of the cow. We add to the discussion.
Reviewer 2: Figure 1: missing error bars
Author: Thanks for your correction. However, the error bars are not missing what happens is that the scale of the graphic did not allow the bars to appear. So, if necessary, we can change the graphic model.
Reviewer 2: What covariate structures were considered?
Author: Thanks for your question. For the most part, the covariate structure that better fitted the data was compound symmetry (CS).